# How stochastic cell fate and endoreduplication yield non-random epidermal patterns

Nicola Trozzi and Mateusz Majda

The Mechanobiology Laboratory, Department of Plant Molecular Biology, University of Lausanne, Switzerland

## Insights

endoreduplication; stochastic cell fate; tissue randomization.

**Corresponding author:**
Mateusz Majda;
Email: mateusz.majda@unil.ch

**Associate Editor:**
Dr. Daphné Autran

## Abstract

Pavement cells in the *Arabidopsis thaliana* epidermis span a wide range of sizes and ploidy levels, but rules that generate this heterogeneity across an organ remain unclear. Clark et al. identify a shared genetic pathway that promotes large, polyploid pavement cells in both sepals and leaves, then ask whether the familiar "scattered" distribution of giant cells is truly random. By combining whole-tissue imaging with two independent computational randomization approaches that regenerate tissues from segmented images while preserving cell size distributions and key boundary constraints, together with a stochastic cell-autonomous model, the authors show how an initially random pattern can later appear clustered relative to a changing random baseline as tissues grow and subdivide. The study provides a quantitative framework for testing spatial organization in cellular mosaics where point-based methods fail, and it shows how proliferation history can convert early stochastic fate decisions into a statistically non-random mature pattern.

The epidermis of a mature *Arabidopsis thaliana* leaf blade can appear uniform at first glance, a sheet of interlocking pavement cells interrupted by stomata and occasional trichomes. Quantitative imaging has long shown, however, that pavement cells vary widely in size (Katagiri et al., 2016; Melaragno et al., 1993). Cell area follows a long-tailed distribution and correlates with endoreduplication, suggesting that a subset of epidermal cells exits the mitotic cycle early and instead continues to grow (Kawade & Tsukaya, 2017). In sepals, these outliers are especially conspicuous. Giant cells can reach high ploidy levels and elongate dramatically, standing out clearly from their neighbours (Roeder et al., 2010). Forward genetic studies have identified a pathway that promotes giant cell fate in sepals (Roeder et al., 2010, 2012) but two questions remained open. Does the same logic shape pavement cell size in leaves, and are giant cell positions random across an organ?

## A shared genetic pathway patterns large epidermal cells in sepals and leaves

In sepals, giant cell fate depends on dose-sensitive activity of *Arabidopsis thaliana* Meristem Layer1 (ATML1), an epidermis-expressed homeodomain transcription factor that acts upstream of Loss of Giant Cells from Organs (LGO), a cyclin-dependent kinase inhibitor that promotes endoreduplication at the expense of division (Roeder et al., 2010; Roeder et al., 2012). Clark et al. (2025) ask whether upstream components of this pathway also influence pavement cell size in leaves. Genetic perturbations reveal a consistent pattern. Loss of function in *Arabidopsis CRINKLY4* (*Acr4*), *Defective KERNEL1* (*Dek1*), *ATML1* or *LGO* reduces the formation of leaf giant cells, whereas *LGO* overexpression increases their number. *ATML1* overexpression does not necessarily increase the number of giant cells per unit area. Instead, it increases the fraction of tissue occupied by giant cells because individual giant cells become exceptionally large.

The organ-level consequences are also informative. Leaves with larger cells do not necessarily grow into larger organs. Across wild type, *atml1-3* and *lgo-2*, overall leaf size is broadly similar, whereas *ATML1* or *LGO* overexpression lines are smaller at maturity (Clark et al., 2025). A consistent interpretation is that mitotic division can compensate for reduced endoreduplication, buffering overall organ size (Horiguchi & Tsukaya, 2011). In sepals, quantitative

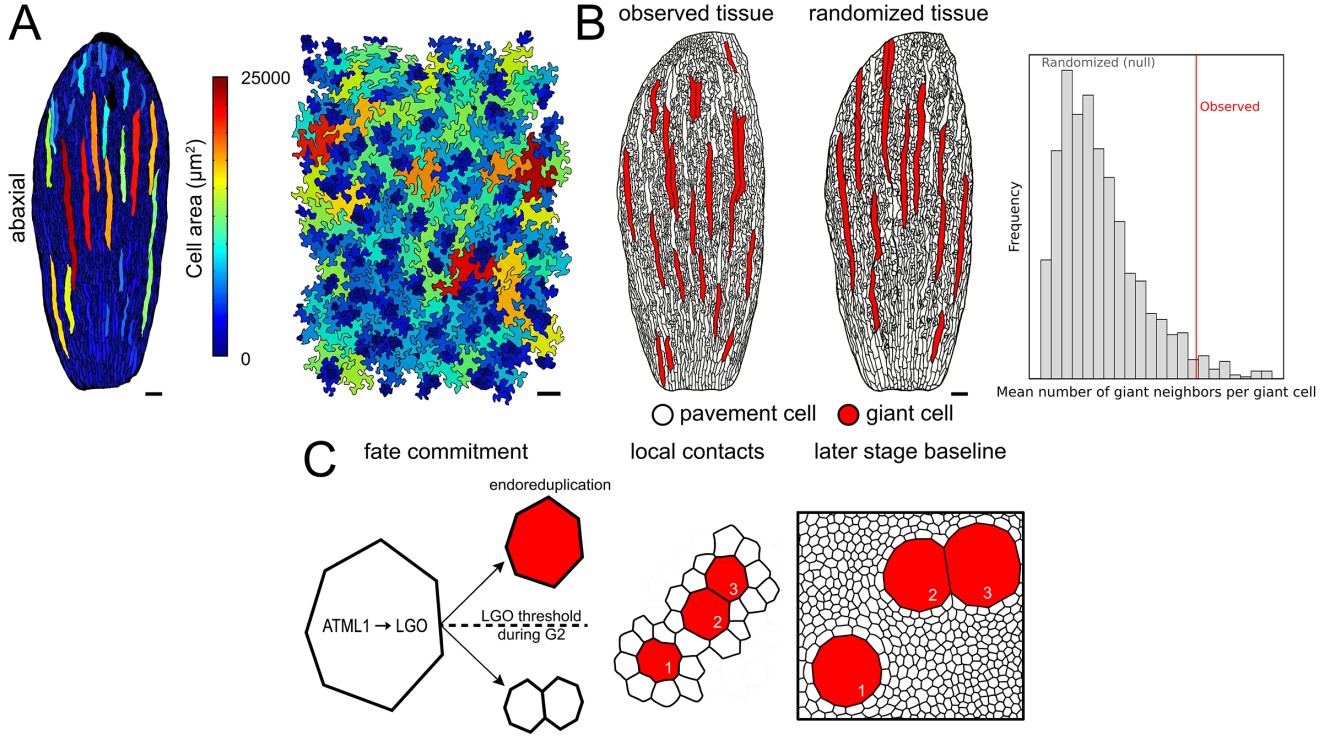

**Figure 1.** From stochastic fate commitment to tissue-scale clustering. (a) Representative epidermal maps illustrating the wide range of pavement cell areas in leaves and sepals, with large cells occupying a disproportionate fraction of tissue area (images adapted from Clark et al., 2025). (b) Statistical quantification of clustering. Left, the observed tissue displays the actual spatial arrangement of giant cells (red). Middle, schematic example of a dmSET-like randomized tissue, where cell positions are shuffled while preserving cell size distribution and boundary constraints. Right, schematic illustration of the randomization test, comparing the observed mean number of giant neighbours per giant cell (red line) with a conceptual null distribution from repeated randomizations (grey bars), to show how apparent clustering can be evaluated against a randomized baseline. Axis values and *p*-values are omitted because the panel is intended to convey the concept rather than report a quantitative analysis. (c) Cell-autonomous model for giant cell formation and apparent clustering. Left, fluctuations in ATML1 activity activate LGO, and an LGO threshold during G2 biases a cell towards endoreduplication (red giant fate) or division (white pavement cells). Middle, local contacts among newly specified giant cells at the time of fate commitment (numbered 1–3). Right, later-stage baseline after tissue growth and cell division. Numbered landmarks (1–3) track the same giant cells to show that while topological contacts are preserved (e.g., between cells 2 and 3), the extensive division of the surrounding pavement cells shifts the null expectation, making the original cluster appear statistically significant relative to the late-stage baseline. Scale bars, 100 μm.

analysis similarly supports robustness of final organ shape to cell size heterogeneity, even when giant cells are more abundant, although the magnitude of shape change can increase in backgrounds with elevated growth variability (Trinh et al., 2024). Leaf shape appears more sensitive than leaf size. Highly anisotropic giant cells in *ATML1* overexpression coincide with more pointed, oblong juvenile leaves, whereas more isotropic giant cells in *LGO* overexpression align with a rounder outline.

## Cell size heterogeneity is a whole-leaf feature, not an abaxial curiosity

A second contribution of the study is descriptive but important. The authors clarify where large pavement cells occur in leaves and how size distributions change during development. Mature rosette leaves show a broad, long-tailed distribution of pavement cell areas. Sepals, by contrast, have many uniformly small pavement cells plus a scattered subset of giant cells, and the largest sepal and leaf cells reach similar maximum areas (Clark et al., 2025) (Figure 1a). Unlike sepals, which restrict giant cells to the abaxial epidermis, leaves contain large pavement cells on both abaxial and adaxial surfaces. This contrast suggests that giant cell formation uses shared machinery, but upstream regulation differs between organs. In sepals, polarity or organ identity cues may gate the *ATML1* to *LGO* module in a surface-specific manner, whereas leaves may deploy

the same module on both sides. Mechanical context could also contribute, since sepals curve and enclose under spatially patterned constraints, whereas leaves expand as a flatter lamina. Timing is a third possibility if the endoreduplication window overlaps differently with proliferation on the two surfaces. Histone reporter fluorescence measured from images can serve as a proxy for nuclear ploidy, and Clark et al. (2025) use that approach to confirm a positive area–DNA content relationship in leaves.

A practical issue underlies these observations. Leaf pavement cells are highly lobed and irregular, so large cells are not always obvious by eye. The authors therefore define leaf giant cells using a size threshold anchored in *atml1-3* tissues, and apply a supervised classifier to separate stomata from pavement cells before labelling giant cells. This pipeline matters because the spatial analyses that follow depend on consistent cell classification across genotypes and developmental stages. With giant cells defined consistently, the next question is whether their apparent scattering reflects random placement or a structured spatial pattern.

## Testing randomness in a cellular lattice requires a null model that preserves cell geometry

The core conceptual challenge is simple to state. Giant cells appear scattered and sometimes contact each other, but are these contacts expected by chance (Figure 1b)? Standard point pattern methods,

which treat cells as points and test randomness using distance-based statistics such as nearest-neighbour spacing, are poorly suited to this problem (Clark & Evans, 1954) because epidermal cells tile the plane, neighbour relationships are explicit and cell sizes and shapes vary strongly (Kuan et al., 2022; Summers et al., 2022). A useful null model must respect that geometry and heterogeneity while breaking any true spatial organization.

To test this, the authors compare observed tissues to randomized tissues generated using dmSET, an image-based shuffling method. dmSET generates randomized tissues from segmented images by shuffling cell positions while preserving the cell size distribution and key boundary constraints, with border cells fixed, producing a null model that respects tissue heterogeneity (Laruelle & Genovesio, 2022; Laruelle et al., 2020). The analysis focuses on a statistic with direct biological meaning, which is the mean number of giant-cell neighbours per giant cell. By building a null distribution from hundreds of randomizations per replicate, the authors test whether the observed value is compatible with random spatial organization, given the heterogeneous cell size distribution of the tissue. In leaves, dmSET can distort cell shape, and the authors used controls to confirm that shape artifacts do not explain the giant-neighbour statistic.

Using this approach, mature sepals and mature leaves show more clustering than expected under the null model. Giant cells have, on average, more giant cell neighbours than randomized tissues predict. The study, therefore, reframes "scattered" as a visual impression that does not rule out statistically detectable clustering.

## When a pattern becomes non-random without changing contacts

One of the most interesting aspects of the study addresses how clustering can emerge over time. Because the plant epidermis lacks intercalation and cell migration, unlike many animal epithelia, it is natural to assume that clustering must be imposed early by an active spacing mechanism (Clark et al., 2025; Zuch et al., 2022). However, it has been shown that a pattern can shift from statistically random to statistically clustered over time without requiring active cell rearrangements, because growth shifts the null baseline (Figure 1c).

The explanation hinges on how the random baseline evolves as the tissue grows. As cell divisions accumulate, the number of possible cellular configurations increases, which shifts the null expectation for neighbour counts. Under this evolving null distribution, high giant-neighbour counts occupy a progressively smaller fraction of the random outcomes. A pattern that falls near the center of the early-stage null distribution can therefore move into the upper tail of the late-stage null distribution when evaluated at later developmental stages. Clark et al. (2025) show in simulations that, as divisions accumulate, a giant-neighbour count that is compatible with randomness early can become statistically incompatible with the later-stage null distribution, even when local neighbour relationships change little. A comparable trend is also observed in time-lapse imaging of sepals (Hervieux et al., 2016).

This point matters beyond giant cells. Late-stage deviations from randomness do not automatically imply late-stage signalling, because proliferation alone can shift the statistical baseline and make early stochastic events appear structured. Leaf age adds another layer to this logic. Juvenile leaves show an early proliferation burst followed by an abrupt decline, whereas adult leaves sustain proliferative growth for longer (Li et al., 2024). Such age-dependent proliferation programmes should shift the null baseline differently across successive leaves, and therefore change when, and how strongly, clustering becomes detectable even if giant cell initiation rules are unchanged.

## A minimal stochastic model links gene-expression noise, cell-cycle decisions and tissue growth

To connect the statistical observations to the mechanism, the authors use a previously published stochastic, cell-autonomous multicellular model (Meyer et al., 2017). In the model, *ATML1* activity fluctuates stochastically and promotes *LGO* activity. Cells progress through the cell cycle with variable timing. After DNA replication, a cell that crosses a threshold level of *LGO* during the G2 phase of the cell cycle does not divide and instead enters an endoreduplication programme, producing a giant cell. Simulations based on this framework reproduce the observed organization, including the gradual emergence of clustering as the tissue proliferates. Notably, the model does not require explicit cell-to-cell communication to generate a non-random tissue-scale outcome.

This modelling choice helps disentangle two ideas that are often conflated in qualitative discussions. Randomness in fate initiation can coexist with non-random spatial organization at later stages once tissue growth is taken into account. This point is made explicit in Clark et al. (2025) by combining tissue-preserving randomizations, time-resolved analysis and a minimal stochastic model that produces late-stage clustering without requiring explicit cell-to-cell communication.

## What to take forward in computational morphodynamics

Two aspects of the work appear broadly applicable to other developing systems. The first is the null model strategy. Any study that assigns cell states on a segmented tissue and asks whether spatial organization differs from chance requires a randomization method that preserves key geometric features that bias neighbour statistics, especially heterogeneous cell sizes and shapes. The dmSET approach provides a practical route to hypothesis testing for patterns that cannot be treated as point processes (Laruelle & Genovesio, 2022).

The second is the time-aware interpretation of pattern metrics. If a statistic compares observed organization to a null distribution, then growth can shift that null distribution even when the observed configuration remains locally stable. Analyses of epidermal mosaics will be stronger if they report developmental trajectories, not only endpoints, because endpoint geometry can reflect earlier growth dynamics and is easier to interpret with temporal context (Trozzi et al., 2026).

Several follow-ups could further sharpen the mechanism. Lineage-resolved live imaging in leaves could test whether local division histories around early giant cells quantitatively predict the later shift in clustering under the null model. Modelling could also incorporate spatial growth gradients, since real leaves develop strong gradients that may bias where the first giant cells arise. The success of the current model indicates that these additions are not required to explain clustering in principle, but added realism could reveal where the minimal explanation stops being sufficient.

Overall, this study links a conserved pathway for endoreduplication-driven cell enlargement to a subtle,

time-dependent patterning phenomenon. The study shows how a mature, statistically clustered arrangement can arise from stochastic initiation combined with tissue growth, and it provides a clear framework for testing similar questions in other cellular mosaics.

**Open peer review.** To view the open peer review materials for this article, please visit http://doi.org/10.1017/qpb.2026.10043.

## Acknowledgements

The authors would like to thank Adrienne H. K. Roeder and Pau Formosa-Jordan for careful reading of the manuscript and for helpful comments.

**Competing interest.** The authors declare none.

**Data availability statement.** No new experimental data or reusable code were generated for this Insight. Figure 1 includes schematic illustrations created for explanatory purposes based on Clark et al. (2025) and does not present a new quantitative analysis.

**Author contributions.** N.T. and M.M. wrote the manuscript.

**Funding statement.** This work was supported by the University of Lausanne and by the Swiss National Science Foundation (SNSF) Starting Grant PS00-3_234905 awarded to MM.

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
