## [Reviewer Report]

Comments to the Author

The manuscript entitled “How stochastic cell fate and endoreduplication yield nonrandom epidermal patterns” discusses the recent paper of Clark et al., 2025 published on PLOS Biology. In my view, this Insight does a good job of summarizing the paper and helping readers understand it better. I would like to support this manuscript for publication in Quantitative Plant Biology, with some minor comments below.

Comments:

- The authors should provide the full name of genes/proteins/mutants on the first mention.

- Lines 51-53 discuss the possible impacts of giant cells on organ shape. I think the paper of Trinh et al., 2024 on Biology Letters touches on this subject and probably can be included in the discussion.

- Sepals and leaves have different distribution of giant cells on the adaxial and adaxial surfaces (line 60). What does it tell us about the shared genetic pathways patterning giant cells in the two organs? Please expand the discussion of possible.

- Line 72: the transition to this part seems abrupt.

- Line 103-104: “A set of giant-cell contacts that looked unremarkable early can therefore become unlikely later when evaluated against the later stage null distribution”. I think the clarity of this sentence can be improved by replacing “looked unremarkable” and “unlikely” by more concrete words.

- Has the observation (random at early stage, non-random at later stage) been described elsewhere? The authors can clarify the novelty of this observation somewhere in the review.

---

## [Reviewer Report]

In this Insight article, Trozzi and Majda provide a critical analysis of Clark, Weissbart and Wang et al. (2025), which investigated how giant cells arise and are spatially patterned in plant aerial organs such as leaves and sepals. The original study reported that common genetic pathways operate to create giant cells in the Arabidopsis leaf epidermis and sepals. Trozzi and Majda offer balanced insights into both the genetic and computational analyses presented in the original study. The final section, “What to take forward in computational morphodynamics,” is particularly compelling, and the broader developmental biology and computational modelling communities may benefit from the perspectives it offers. The manuscript may benefit further from considering the following:

1. Lines 73-76 “Standard point pattern methods are poorly suited to this problem because epidermal cells tile the plane, neighbor relationships are explicit, and cell sizes and shapes vary strongly (Kuan et al., 2022; Summers et al., 2022).”

I think readers will benefit from a brief description of Standard point pattern methods, for example: they are based on distance to nearest neighbours where an individual cell of interest is considered as a point. Also, please cite Clark and Evans, 1954 (https://doi.org/10.2307/1931034)

2. Line 95 “Because plant epithelia rarely rearrange neighbor relationships (Zuch et al., 2022)”

It is not clear what authors mean by plant epithelia rarely rearrange neighbor relationships. It is likely that authors are referring to absence of intercalation/cell migration in plants, unlike animals. Rephrasing the sentence to improve clarity would be helpful. Also, for consistency throughout the text it would be better to use the term epidermis instead of epithelia.

3. Lines 111-112 “Late-stage deviations from randomness do not automatically imply late-stage signaling, because proliferation alone can shift the statistical baseline and make early stochastic events appear structured.”

This is an interesting point, and the authors may consider briefly discussing how differential proliferation patterns associated with leaf age may relate to the emergence and spacing of giant cells in leaf epidermis. The original research article focuses on leaf 1 and 2, where cells undergo an early proliferation burst and then rapidly transition into differentiation. In contrast, later forming leaves exhibit more sustained proliferation, and if and how that could influence giant cell emergence and their spacing patterns remains to be investigated (10.1016/j.cub.2023.12.050).

---

## [Editor Report]

Dear Dr Strozzi and Dr Majda, 

We have now received the reviewers comments on your manuscript. Both are positive, yet asking minor revisions. I fully align with their comments. 

Looking forward to receiving your new version of the manuscript. 

Many thanks for contributing to Quantitative Plant Biology. 

Best regards,

---

## [Reviewer Report]

The authors have addressed all concerns raised previously. The article will be of good value to readers.

---

## [Editor Report]

I thank the authors for addressing all the comments and for their interesting contribution, 

Looking forward to reading the article in QPB,